# Effect of Sodium-Glucose Cotransporter 2 Inhibitors on Clinical and Laboratory Variables in Heart Failure Patients with Reduced Left Ventricular Ejection Fraction in a Latin American Hospital: A Retrospective Study

**Mario Osvaldo Speranza-Sánchez** [1], **José Pablo Díaz-Madriz** [1], **Esteban Zavaleta-Monestel** [1,*], **José Miguel Chaverri-Fernández** [2], **Sebastián Arguedas-Chacón** [1], **Marleny Blanco-Jara** [1], **Abigail Fallas-Mora** [1] and **Luis Daniel Velásquez-Alfaro** [2]

1   Heart Failure Clinic, Hospital Clínica Biblica, San Jose 1307-1000, Costa Rica
2   Department of Pharmacology, Toxicology and Pharmacodependence, University of Costa Rica, San José 2060-1000, Costa Rica
*   Correspondence: ezavaleta@clinicabiblica.com

**Abstract:** Heart failure (HF) is a syndrome suffered by more than 26 million people worldwide. SGLT2 inhibitors are drugs that have been shown to positively affect the management of HF patients, regardless of their diabetes status. A retrospective observational study was conducted on heart failure patients with reduced ejection fraction (HFrEF) enrolled at the HF clinic, who were on SGLT2 inhibitors. For these patients, baseline and follow-up data were collected and analyzed over time. Changes over time were quantified and statistical analysis was conducted to validate whether the changes were significant. After the screening of all the HF program patients, 24 met the inclusion criteria, with an average age of 68 years. Through the study, it was possible to find a statistically significant difference in the values of NT-ProBNP before and after adding a SGLT2 inhibitor in 14 patients ($p = 0.0214$). In addition, there was an improvement in the NYHA functional scale of 71% and no significant change in renal function or other laboratory values. Based on the studied parameters and throughout the clinical changes during the follow-up period, it was possible to establish an improvement in HFrEF patients on SGLT2 inhibitors as part of their therapy.

**Keywords:** sodium-glucose transporter 2 inhibitors; heart failure; natriuretic peptides



## 1. Introduction

Heart failure (HF) is a syndrome suffered by more than 26 million people worldwide. Its impact has been increasing and is expected to continue growing due to the aging population [1]. Projections show that the prevalence of this disease will increase by 46% between 2012 and 2030 [2]. Between 64% and 69% of hospitalizations for cardiovascular (CV) causes in Latin American countries are due to this pathology [3].

Current HF management includes antihypertensive drugs, diuretics, vasodilators, inotropes, sinoatrial node modulators, statins, platelet agglutination inhibitors, anticoagulants, and cardiotonic glycosides, among others [4,5]. Within this wide range of therapies, sodium-glucose cotransporter 2 (SGLT2) inhibitors have been gaining popularity in recent years. These medications are used to treat type 2 diabetes mellitus (DM2), but also have shown a positive effect on the management of HF in diabetic and non-diabetic patients [6].

Various hypotheses have been proposed about the potential mechanisms of benefit of these medications. It has been established that they lead to both osmotic diuresis and natriuresis, as well as lower blood pressure, reduce preload and afterload, generate changes in myocardial metabolism that improve left ventricle systolic function, and cause a decrease in cardiac stress, generating less hypertrophy and fibrosis, reflected in a decrease in cardiac remodeling [7,8]. Animal models and human studies demonstrate that these drugs

cause a shift in myocardial metabolism away from the utilization of energy-inefficient glucose toward the consumption of free fatty acid and ketone bodies, which improve cardiac energetics [9,10], resulting in an improvement in both systolic [11] and diastolic function [12]. Additionally, a sympatholytic effect induced by SGLT2 has been suggested. Through a mechanism that currently has not been fully elucidated, these drugs have the ability to suppress sympathetic nerve activity, similar to beta-blockers, achieving a decrease in heart rates [13,14].

The reduction in the risk of cardiovascular and renal events with SGLT2 inhibitors occurs rapidly after initiation of therapy and persists during treatment, so it is likely that initial diuresis, followed by longer-term changes in tissue management of the sodium, is the basis in terms of reducing the risk of developing or decompensating HF, reducing blood pressure, and preserving renal glomerular function [6].

The evidence supporting the use of SGLT2 inhibitors in patients with heart failure has been provided by different studies. The DAPA-HF study analyzed the behavior of dapagliflozin use in heart failure patients with reduced left ventricular ejection fraction (rLVEF), and it showed that both the risk of disease worsening and the number of deaths from CV causes was lower in patients who received SGLT2i, regardless of the presence or absence of DM2 [15]. Similarly, the DEFINE-HF study showed that dapagliflozin in patients with HF with rEFV is capable of improving the health status of patients, and the concentrations of natriuretic peptide in these, regardless of the presence or absence of DM2 [16]. In addition to the benefit to health status in non-diabetic patients, it has been shown that SGLT2 achieves an improvement in the quality of life of these patients [17].

The EMPEROR-Reduced study analyzed the use of empagliflozin in HF patients with rEFV and showed a reduction in CV deaths and the number of hospitalizations due to the worsening of the disease. This study also demonstrated that this iSGLT2 decreases the deterioration of the glomerular filtration rate (eGFR) over time, resulting in an improvement in renal function [18]. Additionally, in the EMPEROR-Preserved trial, carried out in patients with HF with preserved left ventricular ejection fraction, empagliflozin reduced the combined risk of death from CV causes and hospitalization for worsening disease equally, regardless of the presence or absence of DM2. In this study, a significant reduction was obtained in the total hospitalizations for HF and a modest improvement in the quality of life of the patients at 52 weeks [19,20].

The EMPA-REG OUTCOME study demonstrated a 35% relative risk reduction in hospitalization for HF in patients with DM2 and atherosclerotic vascular disease using empagliflozin. Similarly, in the CANVAS program and the DECLARE-TIMI 58 study, it was possible to confirm the benefit of the SGLT2 inhibitors, canagliflozin, and dapagliflozin, respectively, in hospitalization for HF, death of CV origin, and renal function in the participants [21].

Given the early and continuous reduction in the risk of HF events observed in trials with SGLT2 inhibitors, changes in clinical parameters after initiation of treatment, regardless of NT-proBNP concentration, could be prognostic and could represent determinants key to identifying patients who respond or do not to this therapy [22].

Current evidence suggests that if SGLT2 inhibitors are not contraindicated, they should be first-line therapy for the treatment of all patients with HF since they can have a considerable impact on the prognosis of the disease [7]. Currently, this pharmacological group is incorporated as first-line treatment in the different international guidelines, and the analysis of its efficacy and safety in distinct population groups continues to consolidate these agents as the cornerstone of heart failure therapy globally [23].

Latin America, like the rest of the world, suffers greatly from this pathology. It presents an increase in incidence, hospitalizations, and mortality, and additionally, it is a region that, in general, has fewer health resources for the treatment of this disease and has a great lack of studies carried out in its population [24].

The Heart Failure Program of Hospital Clínica Bíblica (PIC-HCB) consists of a multidisciplinary group of specialists, including cardiologists, geriatricians, nurses, pharmacists,

and rehabilitation specialists, responsible for the assistance and monitoring of patients with heart failure in order to manage the disease and improve the quality of life. Members of this unique program in Central America provide education on the disease and explain the importance and correct use of prescribed medications.

The objective of this study is to perform an analysis of the effect of SGLT2 inhibitors in active patients of the PIC-HCB with rEFV on the clinical and laboratory variables used for the control of heart failure in order to validate the evidence obtained at the local level with other similar clinical studies.

## 2. Materials and Methods

A retrospective observational study of the patients enrolled at the HFC-HCB from January 2018 thru December 2021 was carried out. Patients older than 18 years who started therapy with a lower initial ejection fraction of 40%, and in addition to standard treatment for heart failure had been prescribed an SGLT2 inhibitor for at least 6 months, were included.

For each patient, data were collected to determine the demographic characteristics of the sample, such as age and sex, in addition to the baseline and follow-up values of different diagnostic tests, such as physical examination, vital signs, and NYHA functional class; laboratory tests, such as NT-ProBNP, serum creatinine (sCr), eGFR using the CKD-EPI creatinine equation, and potassium (K+) and sodium (Na+); Kansas (KCCQ) score and Barthel index; adverse effects associated with iSGLT2; the number of hospitalizations; and cardiovascular mortality.

The baseline values of the clinical variables considered for the study were registered. Subsequently, the variables were evaluated every 24 weeks, from the start of treatment with SGLT2 inhibitors, to assess the presence of changes in the results. Statistical analyses were performed to validate whether the changes were statistically significant over time, with a pre-established alpha-type error of less than 5% ($p < 0.05$) and a 95% confidence interval. The Microsoft Excel program and the Statistical Package for the Social Sciences (SPSS in its latest version) were used, and the clinical relevance of the changes was approximated using the Kansas City Cardiomyopathy Questionnaire (KCCQ).

## 3. Results

Of the 47 active patients in the PIC-HCB with a prescription for an SGLT2 inhibitor, only 24 met the inclusion criteria, whose demographic data are presented in Table 1. The mean age of 68 years (35–92 years) was obtained, with an average length of stay in the program of 16 months. Regarding cardiovascular comorbidities regularly associated with heart failure, 75% (18/24) of the patients suffered from arterial hypertension, 46% (11/24) suffered from diabetes mellitus, 17% (4/24) have a history of acute myocardial infarction, and another 17% (4/24) suffer from ischemic heart disease.

Of the 24 selected patients, 18 were prescribed dapagliflozin as an SGLT2 inhibitor at an optimal dose of 10 mg, while the other 6 patients used empagliflozin. In total, 4 empagliflozin patients had been prescribed a dose of 10 mg, while the other 2 had a 25 mg dose. As seen in Table 1, all patients maintained standard treatment of heart failure according to clinical guidelines, which were prescribed before the SGLT2 inhibitor.

After a mean follow-up of 16 months, using a paired Student's *t*-test with a 95% confidence interval, a statistically significant difference was found in the change in NT-ProBNP values for 14 of the 24 selected patients ($p = 0.0124$). The improvement was 57.3% in NT-ProBNP after 6 months of using an SGLT2 inhibitor. These values and the behavior of the other parameters under study are presented in Table 2. Not all the variables consider the whole sample, since it was not possible to obtain the necessary information to perform the respective before-and-after analyses for all the patients.

**Table 1.** Demographic and clinical characteristics of patients; Abbreviations: SD, standard deviation; HF, heart failure.

| Characteristic | Value |
|---|---|
| **Demographic characteristics** | |
| Age, mean years ± SD | 68 ± 13.6 |
| Male, *n* (%) | 13 (54.2) |
| Female, *n* (%) | 11 (45.8) |
| **Comorbidities** | |
| Arterial hypertension, *n* (%) | 18 (75) |
| Diabetes mellitus, *n* (%) | 11 (46) |
| Acute myocardial infarction, *n* (%) | 4 (17) |
| Ischemic heart disease, *n* (%) | 4 (17) |
| **SGLT2 inhibitor prescribed** | |
| Dapagliflozin, *n* (%) | 18 (75) |
| Empagliflozin, *n* (%) | 6 (25) |
| **Concomitant treatment for HF** | |
| Beta-blocker, *n* (%) | 24 (100) |
| Sacubitril/valsartan, *n* (%) | 24 (100) |
| Mineralocorticoid receptor antagonists, *n* (%) | 9 (37.5) |
| Loop diuretics, *n* (%) | 7 (29.2) |
| Ivabradine, *n* (%) | 4 (16.7) |

**Table 2.** Comparison of the main clinical parameters before and after the establishment of the SGLT2 inhibitor. Abbreviations: BUN, blood urea nitrogen; BP, blood pressure.

| Variables | Initial Mean ± SD | Final Mean ± SD | Magnitude of Change * |
|---|---|---|---|
| Heart rate (beats/min) | 70.0 ± 12.8 | 67.0 ± 9.8 | ▼ 4.3 |
| NT-ProBNP (pg/mL) | 3855.0 ± 4095.0 | 1647.0 ± 3661.0 | ▼57.3 |
| Systolic BP (mmHg) | 115.0 ± 20.4 | 119.0 ± 16.4 | ▲ 3.5 |
| Diastolic BP (mmHg) | 69.0 ± 7.6 | 70.0 ± 9.4 | ▲ 1.4 |
| Barthel index (points) | 90.0 ± 9.6 | 93.0 ± 11.9 | ▲ 3.3 |
| KCCQ score (points) | 49.0 ± 14.4 | 59.0 ± 5.9 | ▲ 20.4 |
| Sodium (mEq/L) | 140.1 ± 2.4 | 140.3 ± 2.8 | ▲ 0.1 |
| Potassium (mmol/L) | 4.3 ± 0.8 | 4.5 ± 0.4 | ▲ 4.7 |
| Serum Creatinine (mg/dL) | 1.2 ± 0.4 | 1.2 ± 0.3 | - |
| eGFR (mL/min/1.73 m$^2$) | 59.2 ± 21.7 | 63.7 ± 22.7 | ▲ 7.7 |
| BUN (mg/dL) | 21.0 ± 9.3 | 17.0 ± 3.8 | ▼ 19.3 |

\* ▲: increased value; ▼: decreased value

Table 3 shows the changes in the NYHA functional class. A total of 70.8% of the patients presented improvements in the stage of their functional class of heart failure, 29.17% of the patients did not present improvement, and no patient presented worsening. No adverse events associated with SGLT2 inhibitor therapy were reported during the study, and no hospitalizations or deaths associated with worsening heart failure were reported.

**Table 3.** Variations of NYHA in patients following the addition of SGLT2 therapy.

| Modification of the Functional Class | Number of Patients | Changes |
|---|---|---|
| Demotion of a class | 12 | NYHA III to NYHA II: 9 patients<br>NYHA II to NYHA I: 3 patients |
| Decline of two classes | 5 | NYHA III to NYHA I: 5 patients |
| No modifications | 7 | NYHA I: 3 patientsNYHA II: 4 patients |

Abbreviation: NYHA, New York Heart Association.

## 4. Discussion

The present study sought to analyze the effect of SGLT2 inhibitors on the clinical and laboratory variables used to control heart failure in active patients of the heart failure

program of Hospital Clínica Bíblica, with the goal of validating the data obtained locally with the existing international evidence.

The treatment of heart failure with SGLT2 inhibitors has clinically demonstrated high effectiveness and safety in patients with reduced ejection fraction and with preserved ejection fraction. A reduction in cardiovascular death and the number of hospitalizations has been demonstrated.

The mechanisms of action of its functioning in HF are not entirely clear. Some hypotheses indicate that its effects on blood pressure, myocardial metabolism, and the maintenance of renal function are the cause of the improvement in laboratory values and echocardiographic variables, such as blood pressure, heart rate, serum creatinine, and NT-ProBNP, among others [7,15,25–29].

The NT-ProBNP marker is a biomarker of neurohormonal activation, which is associated with myocardial damage and structural damage. This has an important diagnostic and prognostic value in HF, since a decrease in its values is generally associated with a reduction in morbidity and mortality, hospital readmissions, and worsening of the disease. In this study, the reduction in NT-ProBNP values proved to have statistically significant relevance, clinically confirming the positive effect that SGLT2 inhibitors have on the values of this marker. These positive results could confirm the clinical improvement of the patients, coinciding with the results of other internationally validated studies [6,7,18,29–31].

Regarding other parameters, such as renal function, electrolyte quantification, serum creatinine, and urea nitrogen, no relevant changes were quantified over time, which can be interpreted as the maintenance of renal function, which is an expected result in patients with heart failure using an iSGLT2 [6,22]. Probable subsequent nephroprotection probably requires longer exposure time and a larger sample of patients, as is the case with the changes that are expected for both blood pressure and heart rate.

The results of improvement in the NYHA functional classification can be associated with an improvement in the Kansas and Barthel scores, which together represent prognostic values that allow discriminating or inferring the risk of worsening of the disease in patients with HF and allow predicting the risk of hospitalization and short-term mortality in this population [5,7,20,32,33]. In the results of this research, an improvement in the NYHA functional classification can be observed, with 70.8% of the patients presenting an improvement. In addition, there was an improvement in the Kansas and Barthel indices, which, although not statistically significant, could indicate that after 6 months of treatment with an iSGLT2, the patients had a greater ability to perform their daily functions and presented fewer symptoms typical of HF [7,15,19,26].

The main limitation of this research is associated with the small size of the sample, although we believe it is important to report these findings for different reasons. As mentioned above, in Latin America, there are very few reports related to this pathology and the use of these medications. Additionally, most of these types of patients in the country are managed through the public health system. Other limitations correspond to the time associated with the follow-up of some patients and the difficulties in obtaining complete follow-up data among the patients who met the inclusion criteria. Initially, it was considered to take into account the echocardiographic variables of the patients to fully evaluate the ejection fraction of the left ventricle, but it was not possible to obtain this information from all the digital records of the patients. A considerable number of patients did not have NT-ProBNP values, which made it impossible to calculate the significance of the results for the entire sample, and the lack of ejection fraction follow-up values made it necessary to exclude this value from the study parameters.

Almost half of the patients with diabetes additionally may have played a role in skewing the changes observed towards bigger benefits with the SGLT2 inhibitor, although the literature has shown benefits in patients with and without diabetes. Finally, the ethnicity/race of the study's population lacks diversity, since all the patients are from and reside in Costa Rica. Furthermore, it is beneficial for the reason of few studies carried out in the region.

## 5. Conclusions

For the patients studied and analyzed in the Heart Failure Program of the Hospital Clínica Bíblica, it was possible to collect clinical and laboratory parameters for monitoring over time. This made it possible to observe an improvement in marker variables, such as NT-ProBNP, maintenance of renal function, and improvement in the Kansas and Barthel scores. A decrease in morbidity and mortality and hospital readmissions could then be expected in the future, as well as an improvement in the quality of life of patients. With this, the benefit associated with SGLT2 inhibitors, confirmed in other studies and evidenced in the clinical and laboratory parameters described here, can be confirmed.

**Author Contributions:** Conceptualization, M.O.S.-S., J.P.D.-M., M.B.-J. and J.M.C.-F.; methodology, L.D.V.-A., A.F.-M. and J.M.C.-F.; software, A.F.-M. and L.D.V.-A.; validation, M.O.S.-S., J.P.D.-M., J.M.C.-F. and M.B.-J.; formal analysis, J.P.D.-M., J.M.C.-F. and A.F.-M.; investigation, M.O.S.-S., J.P.D.-M. and J.M.C.-F.; resources, M.O.S.-S., J.P.D.-M. and E.Z.-M.; data curation, A.F.-M. and L.D.V.-A.; writing—original draft preparation, M.O.S.-S., S.A.-C., A.F.-M., L.D.V.-A., J.M.C.-F. and J.P.D.-M.; writing—review and editing, M.O.S.-S., J.P.D.-M., J.M.C.-F., S.A.-C. and E.Z.-M.; visualization, S.A.-C. and E.Z.-M.; supervision, M.O.S.-S.; project administration, M.O.S.-S. and J.P.D.-M.; funding acquisition, E.Z.-M. All authors have read and agreed to the published version of the manuscript.

**Funding:** This research received no external funding.

**Institutional Review Board Statement:** The study was conducted in accordance with the Declaration of Helsinki and approved by the Scientific Ethical Committee of the University of Costa Rica, Costa Rica (approval date 13 June 2022), approval reference number CEC-284-2022.

**Informed Consent Statement:** Patient consent was waived due to the minimal risk to the study population.

**Data Availability Statement:** The data presented in this study are available on request from the corresponding author.

**Acknowledgments:** The authors thank Hospital Clínica Bíblica for the support provided to conduct this research.

**Conflicts of Interest:** The authors declare no conflict of interest.

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
