# Peer review of "Effect of Sodium-Glucose Cotransporter 2 Inhibitors on Clinical and Laboratory Variables in Heart Failure Patients with Reduced Left Ventricular Ejection Fraction in a Latin American Hospital: A Retrospective Study"

_hearts, doi:10.3390/hearts4010003_

Round 1
Reviewer 1 Report
The authors report that initiation of SGL2i in HFrEF outpatients reduce NT-ProBNP and improve quality of life.
The authors are to be congratulated for investigating the effect of SGLT2i on real-life patient in Costa Rica. The results are consistent with the literature, and conclusions are supported by the data.
This peer reviewer raises the following issues:
Major issues:
- The main issue is the small sample size (n=24).
- Authors mention that they will evaluate LVEF (Lune 99-100). The authors should present a Table 3 with all the echo data before and after initiation of SGLT2i: LVEF, LV mass, LV volumes, RV and LA dimensions, IVC diameter
- The authors mention that they will evaluate quality of life as per KCCQ and Barthel (line 101). Both parameters have to be presented in the Results section
Minor issues:
- Table 2: Please provide eGFR and microalbuminuria
- Line 45: The authors write “generate changes in myocardial metabolism that decrease cardiac output”. Actually, SGLT2i “generates changes in myocardial metabolism that improve LV systolic function. The authors should mention that ”SGLT2i cause a shift in myocardial metabolism away from utilization of energy-inefficient glucose toward the consumption of free fatty acid and ketone bodies, which improve cardiac energetics”. This has been demonstrated in animal models (please quote J Am Coll Cardiol. 2019 Apr 23;73(15):1931-1944) and in humans (please quote Circulation. 2022 Sep 13;146(11):819-821), This enhanced energetics cause improvement in both systolic (please quote J Am Coll Cardiol. 2021 Jan 26;77(3):243-255) and diastolic function (please quote JACC Cardiovasc Imaging. 2021 Feb;14(2):393-407).
- Line 47: When discussing that SGLT2i reduce cardiac fibrosis, the authors should quote the pioneering work demonstrating this (JACC Heart Fail. 2021 Aug;9(8):578-589)
- Line 61: The authors should mention that SGLT2i improve quality of life (please quote Diabetes Metab Syndr. 2022 Feb;16(2):102417).
Author Response
First of all, we are very grateful for your feedback. We try to take all comments into account to improve the quality of the manuscript.
Major issues:
- The main issue is the small sample size (n=24).
We agree that the study population is very small. We add the following paragraph within the limitations of the study:
" The main limitation of this research is associated with the small size of the sample, although we believe it is important to report these findings for different reasons. As mentioned above, in Latin America there are very few reports related to this pathology and the use of these medications. Additionally, most of these types of patients in the country are managed through the public health system".
In the manuscript, we try to explain that despite having a small sample, our study is important for the region due to the lack of reports in Latin America.
- Authors mention that they will evaluate LVEF (Lune 99-100). The authors should present a Table 3 with all the echo data before and after initiation of SGLT2i: LVEF, LV mass, LV volumes, RV and LA dimensions, IVC diameter
Another limitation that we had in this study. Due to an error in the final manuscript, we maintained that echocardiographic variables were going to be evaluated, but it was not possible for all the patients, therefore, having excluded more patients who did not have these results would leave us with an even smaller sample. We try to explain ourselves better with the following paragraph
"Initially, it was considered to take into account the echocardiographic variables of the patients to fully evaluate the ejection fraction of the left ventricle, but it was not possible to obtain this information from all the digital records of the patients. A considerable number of patients did not have NT-ProBNP values, which made it impossible to calculate the significance of the results for the entire sample, and the lack of ejection fraction follow-up values made it necessary to exclude this value from the study parameters."
- The authors mention that they will evaluate quality of life as per KCCQ and Barthel (line 101). Both parameters have to be presented in the Results section
The values ​​are already in the table but they have been accommodated and presented in a better way.
Minor issues:
- Table 2: Please provide eGFR and microalbuminuria
We calculate the eGFR and add it to the table but we do not have the values ​​of microalbuminuria
- Line 45: The authors write “generate changes in myocardial metabolism that decrease cardiac output”. Actually, SGLT2i “generates changes in myocardial metabolism that improve LV systolic function. The authors should mention that ”SGLT2i cause a shift in myocardial metabolism away from utilization of energy-inefficient glucose toward the consumption of free fatty acid and ketone bodies, which improve cardiac energetics”. This has been demonstrated in animal models (please quote J Am Coll Cardiol. 2019 Apr 23;73(15):1931-1944) and in humans (please quote Circulation. 2022 Sep 13;146(11):819-821), This enhanced energetics cause improvement in both systolic (please quote J Am Coll Cardiol. 2021 Jan 26;77(3):243-255) and diastolic function (please quote JACC Cardiovasc Imaging. 2021 Feb;14(2):393-407).
- Line 47: When discussing that SGLT2i reduce cardiac fibrosis, the authors should quote the pioneering work demonstrating this (JACC Heart Fail. 2021 Aug;9(8):578-589)
- Line 61: The authors should mention that SGLT2i improve quality of life (please quote Diabetes Metab Syndr. 2022 Feb;16(2):102417).
We appreciate all references supplied. We fix the text and include all the references correctly

Reviewer 2 Report
The present manuscript by Speranza-Sanchez et al. is an interesting retrospective study on the effects of SGLT2 inhibitors on clinical variables and lab values of a small Latin American HFrEF patient population. Despite some major limitations, outlined by the authors themselves, the study is well designed and reports interesting findings. I just have a couple of important comments for the authors to address:
1) The authors have ignored the well documented sympatholytic benefits of SGLT2 inhibitors (e.g., see: PMID: 34299304; PMID: 32340780, two reviews that should be cited) which could very well have played a role in the observed trend towards lower HR in the SGLT2 inhibitor-treated patients.
2) The fact that about half of the patients had diabetes may have played a role in skewing the changes observed towards bigger benefits with the SGLT2 inhibitor. The authors need to acknowledge that as another limitation of their study in "Discussion".
3) Ethnicity/race of the study`s population needs also to be taken into account, upon discussing the study`s limitations.
Author Response
First of all, we are very grateful for your feedback. We try to take all comments into account to improve the quality of the manuscript.
1) The authors have ignored the well documented sympatholytic benefits of SGLT2 inhibitors (e.g., see: PMID: 34299304; PMID: 32340780, two reviews that should be cited) which could very well have played a role in the observed trend towards lower HR in the SGLT2 inhibitor-treated patients.
We appreciate the recommendation and make the corresponding reference
2) The fact that about half of the patients had diabetes may have played a role in skewing the changes observed towards bigger benefits with the SGLT2 inhibitor. The authors need to acknowledge that as another limitation of their study in "Discussion".
3) Ethnicity/race of the study`s population needs also to be taken into account, upon discussing the study`s limitations.
For points 2 and 3 we add the following paragraph in the limitations of the study:
Almost half of the patients with diabetes additionally may have played a role in skewing the changes observed towards bigger benefits with the SGLT2 inhibitor, although the literature has shown benefits in patients with and without diabetes. Finally, the ethnicity/race of the study`s population lacks diversity, since all the patients are from and reside in Costa Rica. Furthermore, it is beneficial for the reason of few studies carried out in the region.
Round 2
Reviewer 2 Report
No further comments.